# Asymptomatic osteolysis as a risk factor for cardiovascular disease after total hip arthroplasty: A retrospective cohort study

Agata Rysinska[1,2]*, Sara Aspberg[1,3,4], Thomas Eisler[1,2], Michael Axenhus[1,2], Nils P. Hailer[5,6], Daniel Hallman[1,7], Evaldas Laurencikas[1], Max Gordon[1,2], Olof Sköldenberg[1,2,6]

1 Karolinska Institutet, Department of Clinical Sciences, Danderyd Hospital, Division of Orthopaedics, Stockholm, Sweden, 2 Danderyd University Hospital, Department of Orthopaedics, Stockholm, Sweden, 3 Norrtälje sjukhus Department of Cardiology, Norrtälje, Sweden, 4 Karolinska Institutet, Department of Clinical Sciences, Danderyd Hospital, Division of Cardiovascular Medicine, Stockholm, Sweden, 5 Section of Orthopaedics, Department of Surgical Sciences, Uppsala University Hospital, Uppsala, Sweden, 6 The Swedish Hip Arthroplasty Register, Gothenburg, Sweden, 7 Danderyd University Hospital, Department of Radiology, Stockholm, Sweden

* agata.rysinska@regionstockholm.se

## Abstract

### Background and purpose

A clinical problem in total hip arthroplasty (THA) is periprosthetic osteolysis/aseptic loosening. There is a possible association between periprosthetic osteolysis and higher risk of cardiovascular disease (CVD). The aim of this study was to investigate whether THA patients with asymptomatic periacetabular osteolysis, have an increased long-term risk of CVD compared to THA patients without osteolysis, assess time to the event, and compare possible cardiovascular risk markers between the two groups.

### Patients and methods

We conducted a retrospective cohort study of 139 patients treated with uncemented THA between 1992 and 2007. All patients were assessed by computed tomography of the affected hip to sort patients in one group with osteolysis and one without. The statistical analysis was based on descriptive statistics. The Kaplan-Meier method was used to estimate time to event and Cox regression models were fitted to calculate crude and adjusted hazard ratios (HR) with 95% confidence intervals (95% CI).

### Results

There were 33 patients with periacetabular osteolysis and 106 patients without. Mean follow-up time was 16.6 years (range, 12–30). 16 patients (11%) were diagnosed with CVD after a mean (SD) of 14 (6) years, with a higher relative incidence 8/33 (24%) vs

**Data availability statement:** Data contain potentially identifying or sensitive patient information. Data can therefore not be shared openly. Data may be available pending a data retrieval request. Researchers may request data from the Stockholm Ethics Committee (contact via registrator@etikprovning.se) for researchers who meet the criteria for access to confidential data.

**Funding:** The author(s) received no specific funding for this work.

**Competing interests:** The authors declare that they have no competing interests.

8/106 (8%) of CVD among patients with periacetabular osteolysis. The hazard ratio was 1.6, 95% confidence interval (0.4–5.8). There were no differences in presence of cardiovascular risk markers or ECG abnormalities.

## Interpretation

We found no association between periacetabular osteolysis and possible risk markers for CVD. There was a trend towards higher incidence of CVD among patients with periacetabular osteolysis. Based on the results from our relatively small cohort it cannot be excluded that low-grade peri-implant inflammation may increase long term risk of CVD following THA.

## Introduction

Primary osteoarthritis is a multifactorial disease with a low-grade inflammatory component [1]. The end-stage surgical treatment for this condition—arthroplasty surgery—can *per se* trigger other inflammatory processes [2]. As the implant ages, wear debris is generated, which can lead to inflammatory induction of bone resorption adjacent to the implant, defined as osteolysis. One of the most common explanations is that development of excess wear particles produces a proinflammatory state [3,4]. It is widely accepted that particulate debris [5] is the main cause of this condition. It is also shown that osteolysis can be induced by mechanical stimulus of fluid pressure [6] The proinflammatory state leads to aggravated osteoclast differentiation and macrophage activation. This process in turn leads to local peri-implant osteolysis and aseptic loosening around the prosthesis components. Computed tomography (CT) of the affected hip is considered gold standard for detection of osteolysis [7–12].

Osteolysis and aseptic loosening are the most common reasons for total hip arthroplasty (THA) revision surgery [13–15]. The result might be extensive revision surgery [14]. Revision surgery due to aseptic loosening/osteolysis is correlated with worse outcomes compared to primary surgery [3,10]. We have previously shown an increased long-term risk of cardiovascular-related mortality in patients treated with THA [16] and an increased relative risk of cerebrovascular events (CVE) among patients with osteoarthritis who received THA and later underwent revision surgery due to loosening of the prosthesis [17]. Similarly, cardiovascular mortality rate after total knee arthroplasty exceeds that in the general population after 10 years [18].

Inflammatory processes are currently considered as central to the development and complications of CVD [19–21]. Numerous biomarkers involved at various levels [22] of the inflammation cascade have been shown to be associated with adverse cardiovascular outcomes [23]. Leukocytes have a significant role in the inflammatory process and the number of leucocytes is recognized as an inflammatory marker and predictor of cardiovascular events [22].

Elevated high sensitivity c-reactive protein (hs-CRP) levels can independently predict risk of all cause, cardiovascular mortality in the general population [23]. Also

elevated serum level of LDL cholesterol (LDL-C) is a well known risk factor for CVD with accumulation in the vessel wall leading to atherosclerotic disease [24].

ECG is a standard routine examination procedure in cardiac medicine [25,26]. The resting ECG is a well-established diagnostic tool for detecting heart disease. Aside from their use in the clinical situation, ECG:s from allegedly healthy subjects have been used to study the prevalence, correlates and the predictive value of asymptomatic heart diseases in the general population [27,28]. The most common findings of ischaemic ECG are ST-segment or T-wave abnormalities, or both (ST-T abnormalities) [25–30]. Many epidemiological studies have concentrated on the association between initial ECG abnormalities and later fatal and non-fatal CVD in a standardized way. A review by Healy [31] have shown that these abnormalities are associated with an increased risk of CVD-both cross-sectionally and prospectively. ECG is a good method for risk stratification of asymptomatic patients given its low cost, wide use and safety. However, the usefulness of ECG in screening of asymptomatic adults is still debatable because clinical implications of ECG abnormalities are unclear in low risk individuals [32].

The aim of this study was to investigate whether THA patients with asymptomatic periacetabular osteolysis, determined by CT, have an increased long-term risk of CVD compared to THA patients without osteolysis, and to assess time to the event.

## Patients and methods

In this retrospective cohort study patients were recruited from consecutive series of patients operated between 1992-01-01 and 2007-01-01 at one of three sites, Danderyd Hospital and Södersjukhuset (in Stockholm, Sweden), and Uppsala University Hospital in Uppsala, Sweden. Eligible patients were recruited through a combination of an in-hospital database search and search in the Swedish Hip Arthroplasty Register (SHAR) [33]. The SHAR is a national quality registry of THA operations in Sweden, allowing for patient-specific follow-up with a capture rate of 97%. The registry is the second oldest arthroplasty quality registry in the world and includes primary surgeries as well as reoperations. Patients treated with an uncemented THA with metal-on-polyethylene articulation performed due to primary osteoarthritis were considered eligible for inclusion. We aimed for a minimum follow-up time of least 10 years due to the slow development of osteolysis. Therefore, patients who underwent surgery up to 2007 were included. Only the first THA was included for patients who underwent bilateral surgery. The exclusion criteria included pain from the hip (Visual Analogue Scale) [VAS] score of ≥3, with 0 being no pain and 10 extreme pain, hip surgery after the primary surgery, and use of bisphosphonates. The pain-VAS cut-off was chosen arbitrarily to possibly reduce implant complications such as chronic infection.

Research nurses contacted eligible patients by letter. If no response a second letter was sent. If still no response a third attempt was made to reach the patient by phone. At the inclusion visit, patients who met the inclusion criteria received both oral and written information and gave their written informed consent to participate in the study.

### Baseline variables

All patients were interviewed by the first author (AR) or (OS) at the inclusion visit according to a pre specified case report form (CRF). Data from each patient's digital medical charts were collected from one year before the primary surgery and until the end of study follow-up (2019-04-11). Diagnoses (ICD-9 and −10 codes) were collected from the electronic patient record of each patient. Surgery dates, specific information on prosthesis components, and the side of surgery were obtained from the SHAR and crosschecked with the digital medical charts. Follow-up data after the primary surgery on cardiovascular diagnoses and causes of death were collected until 2019-04-11, i.e., after a minimum of 12 years. The data were collected in a digital case report using REDCap (Research Electronic Data Capture) tools provided by Karolinska Institutet [34]. Data was entered in the REDCap database by the main author (AR).

Variables collected at baseline were age at primary surgery, sex, presence of hypertension, myocardial infarction or cerebral infarction prior to surgery, and ASA-classification [35]. This classification gives an indication of the patient's health

status and is a common score used internationally by anaesthesiologists for the pre-operative assessment of physical health status. The score is also used as a proxy for pre-operative comorbidities in research studies [35].

Blood samples were taken at the inclusion visit by trained nurses with standardized methodology. Patients were fasting over night when samples were taken. All blood samples were analysed at a fully accredited central laboratory (Karolinska Laboratoriet) according to standard operating procedures. The results of blood samples were collected from the electronic patient record of each patient.

Total leukocytes count and high sensitivity c-reactive protein (hs-CRP) were analysed as markers of inflammation. Triglycerides, total cholesterol and low-density lipids (LDL) were analysed as risk markers for CVD.

## ECG

Standard 12-lead ECG was collected at the inclusion date, or from medical charts. ECG recorded at the inclusion visit was registered in resting supine position using standardized procedures in the cardiology outpatient ward at Danderyd Hospital. Senior cardiologist SA reviewed all ECG.

ECG-alterations were defined according to the modified Minnesota code as left or right bundle branch block, ST-segment elevation or depression of more than 1 mm except for ST-elevation in V2 and V3 where more than 2 mm ST-elevation was required, or T-wave inversion of > 0.1 mV [30].

## Definition of osteolysis

Presence of periacetabular osteolysis was determined at the inclusion visit with a CT-scan of the included hip. The focus was on the periacetabular region. Comparisons with immediate pre- or post-operative plain radiographs from the index surgery were made. The definition of periacetabular osteolysis in this project was based on CT scans with a sclerotic, demarcated, periacetabular bone defect without trabeculae in direct connection with the surface of the prosthesis. No classification of different types of osteolysis was made. The periacetabular lesion should be at least 2 cm$^3$.

Patients were assessed using three different General Electric Medical Systems CT scanners. A metal artefact reduction program was used. Tomographic images were set at 3 mm in thickness. The images were reviewed using the IntelliSpacePACS Radiology imaging system, version 4.4.516.1 (Philips Healthcare Informatics, Germany). All reviews were performed on high-resolution screens (BARCO MDCC-6130). Vitrea® Advanced (Vital Toshiba Medical Systems, Stockholm, Sweden) was used for semi-automatic volume determination and for additional 2D and 3D reconstructions. Two independent radiologists performed the measurements – first, a resident in radiology (DH), and later, a senior consultant radiologist (EL). Based on radiographic data patients were divided into two groups, with and without osteolysis.

## Outcome

The outcome variables were diagnoses indicating CVD including cardiovascular events (CVE) such as acute myocardial infarction and development of atherosclerotic disease, according to ICD-9 and −10 codes (Appendix), and the time in years to CVD onset after surgery. Patients with outcome events were censored after CVD onset to avoid similar events appearing several times.

## Statistics

Descriptive statistics are given as counts, percentages and means. The chi-square test and Fisher's exact test were used for comparisons between categorical variables. Student's T-test was used for comparisons between means when means were normally distributed, Mann-Whitney U-test was used for non-normally distributed variables. The level of significance was set at $p < 0.05$ in all analyses. A Kaplan-Meier survival analysis was used to estimate the time to event, with the log-rank test pooled over strata comparing all factor levels in a single test

to test the equality of the survival curves between the groups. A Cox regression model was fitted to calculate unadjusted and adjusted hazard ratios (HR) with 95% confidence intervals (95% CI), with the exposure variable (osteolysis versus no osteolysis). Adjustments were made for relevant confounders including age, sex, or CVD prior to surgery. The ASA classification was left out of the final model to avoid over-fitting. The follow-up time was measured in years from the time of the index THA to any CVE, death or the end of the study on 2019-04-11. The assumption of proportionality of hazards was investigated by plotting unadjusted survival curves for each specific group for exposed and unexposed individuals. All analyses were performed using SPSS 25.0 (IBM Corp., Armonk, New York).

### Ethics

The study was conducted in accordance with the Helsinki [36] declaration and the guidelines of the STROBE criteria [37] for observational cohorts was followed when reporting. Ethical approval was obtained by the Regional Ethics Committee in Stockholm (DNR:2014/1080-32).

## Results

### Patient flow

A total of 139 patients (49.6% females) were included in the study, with a mean age at primary surgery of 53 (SD, 8) years (Fig 1). The mean follow-up time was 16.6 years (range, 12–30). The study sample consisted of 33 patients with and 106 patients without osteolysis. The baseline demographics of the subjects were similar, but the patients in the osteolysis group were slightly younger at primary surgery than those in the control group, and a higher proportion of patients without osteolysis were classified as ASA class 3. The proportion of patients with previous myocardial infarction was higher in the osteolysis group. Previous hypertension, atrial fibrillation and any previous CVE were relatively more common in group without osteolysis. There were no previous cerebral infarctions in any of the groups. All patients in the osteolysis group were operated on before the introduction of highly cross-linked polyethylene (HXLPE). This method was used in 43% of the patients without osteolysis (Table 1).

### CVE outcome

Of all study participants, 16 (11.5%) either suffered a CVE or were diagnosed with atherosclerotic disease after a mean (SD) of 14 (6) years. There were 8 events in each group, with a higher proportion of 24% (8/33) in the osteolysis group than in the group of patients without osteolysis (7.5%, 8/123, p = 0.024). However, the osteolysis group had a significantly longer follow-up time and time to event and a statistically insignificant log-rank test result in the survival analysis compared to the group with osteolysis (Table 2), (Fig 2).The unadjusted risk for developing a CVD was also higher, but not statistically significantly so, with an unadjusted HR of 1.4 (95% CI, 0.4–4.2) vs. an adjusted HR of 1.6 (95% CI, 0.5–5.8). There was no association between outcome and age at primary surgery or pre-operative CVD. The female sex was borderline protective against CVE and atherosclerotic disease in the unadjusted analysis but lost statistical significance after adjustments (Table 3). The data with few patients and outcome events did not allow for testing of interaction effects between osteolysis and sex.

### Markers of inflammation and ECG

Complete ECG and blood specimen were retrieved for 108 out of the 139 patients. Hs-CRP levels were slightly lower in the osteolysis group, but this finding did not reach statistical significance. Leukocyte count and lipid levels did not differ between the groups (Table 4), (Fig 3). Further, there were no differences in presence of ECG abnormalities between patients with and without osteolysis (Table 5).

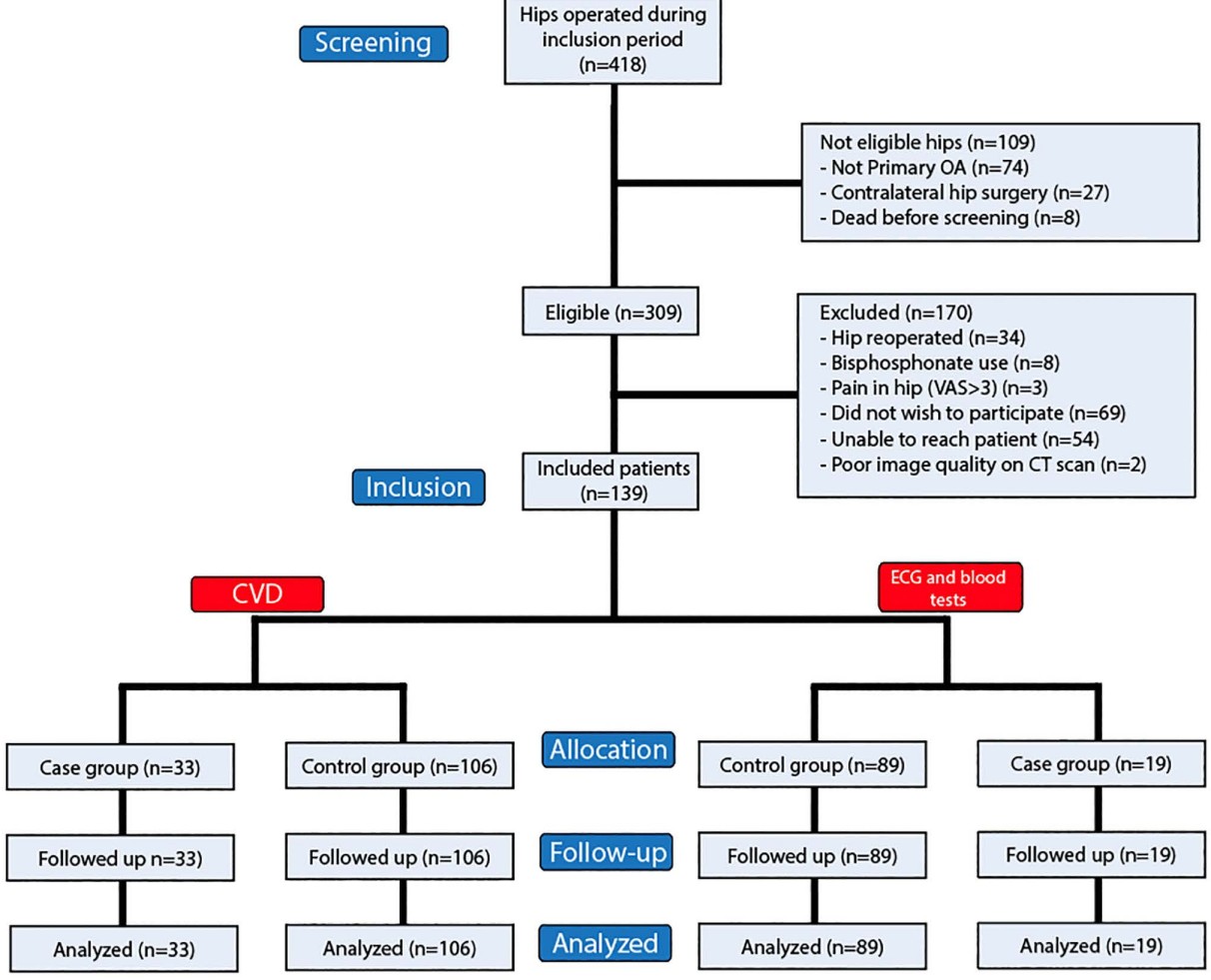

**Fig 1. Flow of patients in the study.** Two patients were excluded due to poor image quality as the presence or absence of osteolysis could not be determined.

## Discussion

We have previously published two large, register based cohort studies on the topic of cardiovascular risk factors associated with THA. The first study showed a higher risk for cardiovascular morbidity and mortality in patients with THA compared to patients without THA eight years and longer after index surgery [16].The second showed patients undergoing revision surgery due to osteolysis having a higher risk for CVD compared to patients without revision surgery [17]. Periacetabular osteolysis is difficult to study due to its "silent" nature without initial symptoms such as pain, and it takes many years to develop, if ever. There is a common denominator for periacetabular osteolysis and atherosclerosis, which is currently suggested as a long time ongoing low-grade inflammation [19,38]. In this study we wanted to examine the association between osteolysis and CVE and risk markers in a clinical setting on an individual level.

Our study did not discern significant discrepancies in cardiovascular risk markers and comorbidities between groups. Notably, levels of inflammatory markers and the prevalence of hypertension or CVD were comparable, except for a tentative difference observed in atrial fibrillation, a condition associated with inflammation [39]. However, the absence

**Table 1. Baseline demographics of the study cohort. P-values derived from Student's T-test and Fisher's exact test.**

| | No osteolysis (n = 106) | Osteolysis (n = 33) | P-value |
|---|---|---|---|
| Age at surgery, y (SD) | 54.0 (8.1) | 50.3 (6.0) | 0.04 |
| Sex | | | 0.11 |
| Female, n (%) | 57 (53.8%) | 12 (36.4%) | |
| Male, n (%) | 49 (46.2%) | 21 (63.6%) | |
| Pre-op hypertension | | | 1.00 |
| No, n (%) | 89 (84.0%) | 28 (84.8%) | |
| Yes, n (%) | 17 (16.0%) | 5 (15.2%) | |
| Pre-op coronary artery disease, n (%)[a] | | | |
| No, n (%) | 94 (88.7%) | 31 (93.9%) | 0.52 |
| Yes, n (%) | 12 (11.3%) | 2 (6.1%) | |
| Pre-op cerebral infarction, n (%) | | | 1.0 |
| No, n (%) | 103 (97.2%) | 33 (100.0%) | |
| Yes, n (%) | 3 (2.8%) | 0 (0.0%) | |
| Any pre-op cardiovascular diagnosis, n (%) | | | 0.52 |
| No, n (%) | 93 (87.7%) | 31 (93.9%) | |
| Yes, n (%) | 13 (12.3%) | 2 (6.1%) | |
| ASA classification | | | 0.03 |
| ASA 1, n (%) | 37 (34.9%) | 7 (21.2%) | |
| ASA 2, n (%) | 46 (43.4%) | 23 (59.9%) | |
| ASA 3, n (%) | 23 (21.7%) | 3 (9.1%) | |
| Polyethylene type | | | <0.001 |
| Standard PE, n (%) | 60 (56.6%) | 33 (100%) | |
| Highly cross-linked PE, n (%) | 46 (43.4%) | 0 (0%) | |
| Stem type | | | 0.16 |
| Bi-Metric | 54 (50.9%) | 22 (66.7%) | |
| CLS | 52 (49.1%) | 11 (33.3%) | |

[a] Includes angina pectoris and myocardial infarction.

**Table 2. Follow-up times, outcomes and time to events. Continuous variables are presented as mean with standard deviation. P-values are derived from Student's T-test and the chi-square test.**

| | No osteolysis (n = 106) | Osteolysis (n = 33) | p-value |
|---|---|---|---|
| Total follow-up time (y) | 15.3 (2.3) | 20.8 (3.6) | < 0.001 |
| Follow-up time to event, death or end of study (y) | 14.9 (3.0) | 20.0 (3.9) | < 0.001 |
| Cardiovascular diagnoses | 8 (7.5%) | 8 (24.2%) | 0.024 |
| Hypertension | 1 | | |
| Angina pectoris | 3 | 4 | |
| Myocardial infarction | 1 | 2 | |
| Atherosclerotic heart disease | | 2 | |
| Chronic ischaemic heart disease | 1 | | |
| Transitory cerebral ischaemic attack | 2 | | |
| Time to event (y) (SD) | 9.1 (4.5) | 17.0 (4.2) | 0.003 |

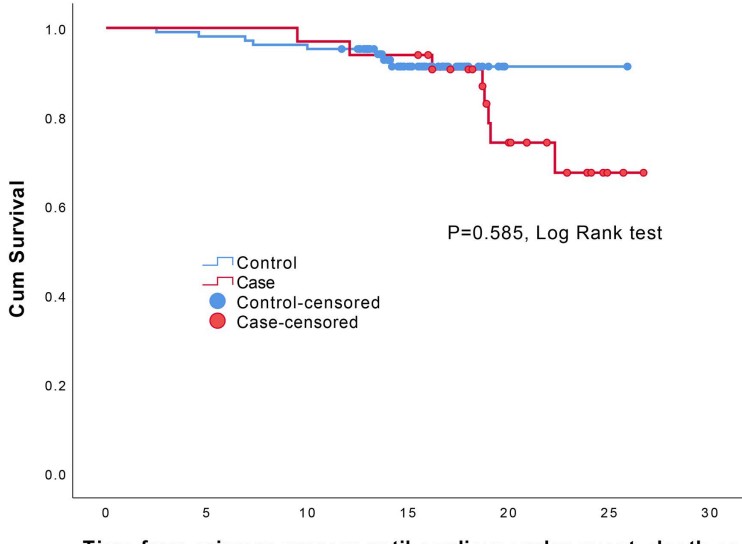

| | Number at risk | | | | | |
|---|---|---|---|---|---|---|
| No | 106 | 104 | 102 | 50 | 1 | 1 |
| Osteolysis | 33 | 33 | 32 | 31 | 17 | 2 |

Kaplan-Meier survival curve with cardiovascular diagnoses as the end point.

**Fig 2. Kaplan-Meier survival curve with cardiovascular diagnoses as the end point.**

**Table 3. Crude and adjusted Cox regression models for the main outcome of cardiovascular diagnoses. The results are presented as hazard ratios (HRs) with 95% confidence intervals (CIs).**

| | | | Crude | | Adjusted | | |
|---|---|---|---|---|---|---|---|
| Variable | Total | Event | HR | 95% CI | HR | 95% CI | P-value |
| Osteolysis | | | | | | | |
| No | 106 | 8 (7.5%) | 1 | | 1 | | |
| Yes | 33 | 8 (24.2%) | 1.36 | 0.44–4.19 | 1.62 | 0.45–5.8 | 0.46 |
| Age | | | 1.05 | 0.98–1.13 | 1.05 | 0.97–1.14 | 0.19 |
| Sex | | | | | | | |
| Male | 70 | 13 (18.6%) | 1 | | 1 | | |
| Female | 69 | 3 (4.3%) | 0.27 | 0.08–0.97 | 0.30 | 0.08–1.07 | 0.063 |
| Pre-op cardiovascular diagnosis | | | | | | | |
| No | 125 | 13 (10.5%) | 1 | | 1 | | |
| Yes | 13 | 3 (10.0%) | 3.18 | 0.85 −11.90 | 2.48 | 0.62–9.95 | 0.2 |

of notable differences in this study does not conclusively negate the possibility of osteolysis serving as a causal factor for CVD as the chronic low-grade inflammation in osteolysis could hypothetically contribute to development of atherosclerosis.

**Table 4. Levels of inflammation markers and lipids.**

| Variable | No osteolysis n=89 | Osteolysis n=19 | p-value |
|---|---|---|---|
| hs-CRP, mg/L (SD) | 2.6 (2.9) | 1.7 (1.5) | 0.56[1] |
| Leukocytes, $10^9$/L (SD) | 6.0 (1.4) | 6.0 (1.4) | 0.77[1] |
| Total cholesterol, mmol/L (SD) | 5.3 (1.1) | 5.0 (1.1) | 0.51[1] |
| LDL, mmol/L (SD) | 3.2 (1.0) | 2.9 (0.9) | 0.28[2] |
| Triglycerides, mmol/L (SD) | 1.2 (0.5) | 1.24 (0.76) | 0.86[2] |

[1] Mann-Whitney U-test.

[2] Independent t test.

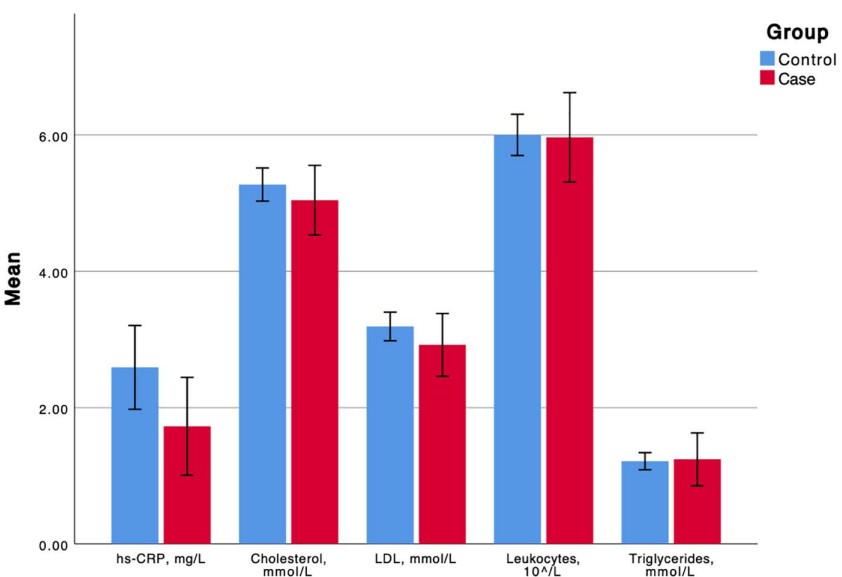

**Fig 3. The mean (95% confidence interval) of serum markers.**

**Table 5. ECG abnormalities.**

| Variable | No osteolysis n=89 | Osteolysis n=19 | p-value |
|---|---|---|---|
| Left sided bundle branch block | 2 | 0 | 1 |
| Right sided bundle branch block | 5 | 0 | 0.58 |
| ST-depression | 3 | 1 | 0.54 |
| ST-elevation | 3 | 1 | 0.54 |
| Pathological Q-wave | 2 | 0 | 1.0 |
| T-wave changes | 9 | 3 | 0.44 |
| Sum (%) | 24 (27) | 5 (26) | |

## Limitations

A major limitation of the study is our limited sample size and the single baseline measurement of hs-CRP levels, leukocytes, and lipids making it difficult to make casual inferences. Repeated measurements had given a more precise

description of the patients. Neither did we control for the use of statins, a class of drugs clinically used to reduce serum lipid levels [40] and the risk of CVD [41]. Statins also suppress osteoclastic bone resorption and has been associated with beneficial effects on bone metabolism and inflammation [42]. Furthermore, unaccounted confounding variables, including lifestyle factors and unmeasured comorbidities, might have impacted the observed outcomes.. Prevalence-incidence bias (Neyman bias) may also have occurred, since patients who have died from CVD before the study make the risk burden of cardiovascular risk factors less severe. ECG might today not be the best reliable predictor for future cardiovascular events, at the time our hospital did not yet perform calcium score computed tomography. The blood tests were collected once-missing potential variations-which is an inherent limitation of a cross-sectional study. Lifestyle factors, comorbidities, or other medications that may affect cardiovascular outcomes are not always mentioned in medical records which could affect the results.

After adjusting for different follow-up times and time to event between groups, plus available confounders, the adjusted HR of 1.6 turned out to be statistically not significant. There was a problem of mismatched observation time, clearly displayed in Fig 2, implying less time for the control group to develop CVD. Age and previously diagnosed CVD are well-established risk factors for subsequent CVE. Of note, the increase in risk estimate associated with osteolysis was considerably higher than the risk associated with age at primary surgery.

Surgical procedures were performed by different surgeons at various times and under different conditions, individual surgical errors or difference in technique could have influenced the outcomes.

Our study is to the best of our knowledge, the first study to investigate the incidence of CVD in patients with or without asymptomatic osteolysis. The connection is biologically interesting, since these conditions seem to share inflammatory pathways, particularly in bone resorption and vascular calcification involving the RANK/RANKL/osteoprotegerin signalling pathway [43]. CVD is currently considered to be mainly driven by inflammation in the vessel wall [19–21] and calcification of the arterial media layer is partly mediated by this pathway. Thus, it would be of interest to know if the benefits of a well-functioning implant with time can be partly offset by a low-grade inflammation derived by wear-debris-induced osteolysis and its possible associated cardiovascular risk. Perhaps, some patients with multiple artificial joints might even benefit from improved cardiovascular surveillance. Obviously, the current study could not determine the existence of such a complex relationship, but we employed strict patient inclusion criteria. There was a high degree of homogeneity originating from validated high-quality national registers combined with clinical examination of each patient and all medical journals were systematically scrutinized.

Our cohort accrued only 16 cardiovascular events, which adequate power only to detect very large effects. Consequently, the study was underpowered to confirm or exclude a modest association. Under realistic event-rate scenarios (12–20%) and an exposure prevalence of 24%, a definitive study would require roughly 1,000–1,600 participants to achieve 80% power. We therefore explicitly acknowledge the alternative interpretation that asymptomatic osteolysis may not be associated with CVD; our data do not refute this possibility. Should the association of CVD and osteolysis be proven, surgeons should take care to include and optimize specific follow up surveillance protocols regarding CVD in these patient populations. However, these protocols should be the subject of interdisciplinary collaboration between orthopaedic surgeons, cardiologists and radiologists. Future, larger multicenter cohorts or registry-based designs with standardized imaging to ascertain osteolysis) will likely be necessary to provide a conclusive answer.

A few patients had to be excluded due to insufficient quality of the tomographic images and some were excluded due to unexplained hip pain. Several attempts have been made to radiographically define and classify osteolysis [7–9,11,12] into a more biological perspective, i.e., with the scope of separating more aggressive lysis from indolent types. For this study, the hypothesis was however that volume of the lesion correlated to its hypothetically systemic effect. Of course, we have no evidence to support this idea and can only speculate on the lesions' biological activity.

From an implant perspective, all articulations had 28 mm cobalt-chrome heads, and the acetabular component was a well-documented [44] uncemented titanium-alloy cup (Trilogy®, Zimmer/Biomet) After 2005, all acetabular liners were

made of highly crosslinked polyethylene, and quite coincidentally, all cases and 57% of the controls turned out to have had a conventional (non-crosslinked) polyethylene liner inserted. We do not know however, how many of the acetabular components associated with osteolysis were supplied with empty screw-holes. The femoral trunnion may of course also have contributed with metallic debris to the inflammatory reaction. In fact, several Bi-Metric® (Biomet) stems had been used in the study cohort; a stem we know is linked to severe trunnionosis [45].

## Conclusion

We found that the presence of asymptomatic periacetabular osteolysis in THA patients was associated with a slightly higher relative incidence of CVD, but the attained risk estimates were statistically non-significant, both before and after adjustment for potential confounders. There was no significant difference in cardiovascular risk markers between the groups. The study suffered lack of statistical power, and it cannot be excluded that low-grade peri-implant inflammation may add to increased long term risk of CVD following THA.

This study underscores the importance of continued inquiry into the intertwined conditions of asymptomatic osteolysis and CVD and their potential clinical implications. Larger sample size and more comparable length in time after surgery may be recommended in new studies.

## Author contributions

**Conceptualization:** Sara Aspberg, Thomas Eisler, Max Gordon, Olof Sköldenberg.

**Data curation:** Michael Axenhus, Max Gordon.

**Formal analysis:** Sara Aspberg, Nils P. Hailer, Max Gordon, Olof Sköldenberg.

**Investigation:** Sara Aspberg.

**Methodology:** Daniel Hallman, Evaldas Laurencikas, Max Gordon, Olof Sköldenberg.

**Resources:** Olof Sköldenberg.

**Software:** Sara Aspberg, Daniel Hallman, Evaldas Laurencikas.

**Supervision:** Thomas Eisler, Nils P. Hailer, Max Gordon, Olof Sköldenberg.

**Validation:** Sara Aspberg, Thomas Eisler.

**Visualization:** Michael Axenhus.

**Writing – original draft:** Agata Rysinska.

**Writing – review & editing:** Sara Aspberg, Michael Axenhus, Nils P. Hailer, Olof Sköldenberg.

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
