## [Decision Letter · Decision Letter 0]

4 Apr 2025

Dear Dr. Rysinska,

Thank you for submitting your manuscript to PLOS ONE. After careful consideration, we feel that it has merit but does not fully meet PLOS ONE’s publication criteria as it currently stands. Therefore, we invite you to submit a revised version of the manuscript that addresses the points raised during the review process.

We look forward to receiving your revised manuscript.

Kind regards,

Nan Jiang

Academic Editor

PLOS ONE

2. Thank you for stating the following in your Competing Interests section:  [No].

Reviewers' comments:

Reviewer's Responses to Questions

**Comments to the Author**

1. Is the manuscript technically sound, and do the data support the conclusions?

Reviewer #1: Yes

Reviewer #2: Yes

2. Has the statistical analysis been performed appropriately and rigorously?

Reviewer #1: Yes

Reviewer #2: Yes

3. Have the authors made all data underlying the findings in their manuscript fully available?

Reviewer #1: No

Reviewer #2: Yes

4. Is the manuscript presented in an intelligible fashion and written in standard English?

Reviewer #1: Yes

Reviewer #2: Yes

Reviewer #1: Dear Dr. Rysinska,

Thank you for publishing your insightful study, "Asymptomatic Osteolysis as a Risk Factor for Cardiovascular Disease." I have previously read your work and find this study to be a compelling and logical extension of your earlier findings. It offers valuable perspectives on the potential link between bone health and cardiovascular risk, contributing to a deeper understanding of this important topic.

I have a question regarding the variability in surgical procedures. Since these surgeries were performed by different surgeons at various times and under different conditions, how did you account for potential individual surgical errors or differences in technique? Could this variability have influenced the outcomes, and if so, how was it addressed in your analysis?

Additionally, I would like to inquire about the use of Cox regression analysis in your study. Given that only 16 patients (11%) were diagnosed with cardiovascular disease (CVD), the number of events is quite low, which may affect the reliability and statistical power of the Cox model. The wide confidence interval (HR: 1.6, 95% CI: 0.4–5.8) indicates a high level of uncertainty in the effect estimate. Considering the limited number of events, did you assess the adequacy of the Cox model for this dataset? Would alternative statistical methods, such as Poisson regression or Firth’s penalized Cox regression, be more appropriate to enhance the robustness of the estimates?

I look forward to your response.

Best regards,

Dr.Farzan Azodi

Reviewer #2: The study addresses an important clinical question about the possible association between periacetabular osteolysis and cardiovascular disease (CVD).The use of the Swedish Hip Arthroplasty Register (SHAR), a well-established national registry with a high capture rate, adds credibility to the dataset. The study carefully selected patients with primary osteoarthritis treated with uncemented THA, ensuring a homogeneous sample. CT scans were used for osteolysis detection, and ECG/blood markers were analyzed for CVD assessment, providing a comprehensive approach.

Patients who died of CVD before study inclusion were not accounted for, possibly underestimating the actual cardiovascular risk. This can be added if authors agree. While ECG was used for cardiovascular assessment, it is not the most reliable predictor of future cardiovascular events. More advanced cardiac imaging or functional tests could have provided stronger evidence.

Needs a plausible explanation. The study only measured hs-CRP, leukocytes, and lipid levels once, missing potential variations over time. It’s important limitation. The study does not fully control confounders for lifestyle factors, comorbidities, or other medications that may affect cardiovascular outcomes.

**Do you want your identity to be public for this peer review?** For information about this choice, including consent withdrawal, please see our Privacy Policy

Reviewer #1: **Yes: ** Farzan Azodi

Reviewer #2: **Yes: ** Syed Muhammad Azfar

---

## [Author Response · Author response to Decision Letter 1]

23 May 2025

May 17, 2025

Agata Rysinska

Manuscript PONE-D-24-55835 Response to Reviewer

Dear Nan Jiang,

Thank you for giving us the opportunity to submit a revised draft of the “Asymptomatic Osteolysis as a Risk Factor for Cardiovascular Disease After Total Hip Arthroplasty:

A retrospective cohort study” manuscript for publication in the PLOS ONE. We appreciate the time and effort that you and the reviewers dedicated to providing feedback on our manuscript and are grateful for the insightful comments on and valuable improvements to our paper. Please see below our comments and answers.

1:

All raw data is uploaded as Supporting Information files

2:

Regarding variability in surgical procedures since the surgeries were performed by different surgeons at various times and under different conditions-if potential individual surgical errors or difference I technique could have influenced the outcomes.

Answer: Thank you for this relevant question.

We have not the surgery reports from the cases.

A total hip arthroplasty is for an orthopedic surgeon a routine surgery, and a surgeon under training is not leaved to perform surgery alone until he or she is assessed by a more experienced surgeon to be ready to do so. It might though influence the results.

3:

Considering the limited number of events, is the Cox model adequate for this dataset?

Answer: You have raised an important question

Due to the nature of the dataset we believe that another model would not change the conclusion of our study.

4:

Patients who died before the study inclusion were not accounted for possibly underestimating the actual cardiovascular risk. ECG was used for cardiovascular assessment-it is not the most reliable predictor of future cardiovascular events. More advanced cardiac imaging or functional tests could have provided stronger evidence. The study measured hs-CRP, leukocytes, and lipid levels once, missing potential variations over time. The study does not fully control confounders for lifestyle factors, comorbidities, or other medications that may affect cardiovascular outcomes.

Answer: We agree with your assessment.

We have not adjusted for patients that died before the study-this could underestimate the risk of death due to cardiovascular causes. ECG might today not be the best reliable predictor for future cardiovascular events, but is readily available and entails no risks. Methods like stress echocardiography, myocardial scintigraphy or computed tomography of the coronary arteries might be used in future studies. The blood tests were collected once-missing potential variations-which is an inherent limitation of a cross-sectional study. Lifestyle factors or comorbidities that may affect cardiovascular outcomes are not always mentioned in medical records which could affect the results. We have, however, included information on the prespecified variables whenever available.

Sincerely

Agata Rysinska

---

## [Decision Letter · Decision Letter 1]

25 Jul 2025

Dear Dr. Rysinska,

Thank you for submitting your manuscript to PLOS ONE. After careful consideration, we feel that it has merit but does not fully meet PLOS ONE’s publication criteria as it currently stands. Therefore, we invite you to submit a revised version of the manuscript that addresses the points raised during the review process.

We look forward to receiving your revised manuscript.

Kind regards,

Nan Jiang

Academic Editor

PLOS ONE

Journal Requirements:

Reviewers' comments:

Reviewer's Responses to Questions

**Comments to the Author**

Reviewer #3: (No Response)

Reviewer #4: (No Response)

2. Is the manuscript technically sound, and do the data support the conclusions?

Reviewer #3: Yes

Reviewer #4: Partly

3. Has the statistical analysis been performed appropriately and rigorously?

Reviewer #3: Yes

Reviewer #4: I Don't Know

4. Have the authors made all data underlying the findings in their manuscript fully available?

Reviewer #3: Yes

Reviewer #4: Yes

5. Is the manuscript presented in an intelligible fashion and written in standard English?

Reviewer #3: Yes

Reviewer #4: Yes

Reviewer #3: I am honored to have the opportunity to review this first study investigating the incidence of CVD in patients with or without asymptomatic osteolysis.

1. Inclusion Criteria for Study Population

The manuscript states that patients treated with uncemented THA with metal-on-polyethylene articulation were considered eligible for inclusion. Were ceramic-on-polyethylene and other bearing combinations excluded? If so, what was the rationale for this exclusion? Similarly, what was the reasoning behind excluding cemented cups and cemented stems?

2. Exclusion of Bisphosphonate Users

Could you please clarify the rationale for excluding patients who were using bisphosphonates?

3. Baseline Characteristics and Statistical Adjustment

The results indicate that "the proportion of patients with previous myocardial infarction was higher in the osteolysis group." Was this baseline imbalance ultimately adjusted for in the final analysis? Is the adjusted hazard ratio of 1.6 (95% CI, 0.5-5.8) the value after adjusting for this and other confounders? I apologize for my limited comprehension, but I would appreciate clarification on this point.

4. Sample Size and Study Conclusions

The study found no significant differences between groups in inflammatory markers (hs-CRP, leukocyte count), lipid levels, or frequency of ECG abnormalities, and the authors attributed this to insufficient sample size. If this is indeed the case, should publication be deferred until a larger sample size can be obtained?

I believe there is a substantial possibility that the true conclusion might be that asymptomatic osteolysis is not associated with CVD development. This alternative interpretation should be acknowledged and discussed. Furthermore, I suggest that the value of this paper would be enhanced by addressing how large a sample size would be needed to definitively conclude "no association" and providing your expert opinion on this matter.

Such discussion would provide important guidance for future research design and help readers better interpret the clinical significance of these findings.

Reviewer #4: I appreciate the effort made by the authors in this interesting study

As an observational retrospective study there are some flaws

1- The definite association between periprosthetic osteolysis and the resulted inflammatory response that causes CVS and cardiac disease has not been proven.

2- The multifactorial etiology of cardiovascular disease makes an obvious difficulty of establishing correlation between osteolysis and CVD.

3- It is very difficult to find matched patients’ characteristics in each group with many variables that will affect the result.

4- The small number of patients studied is another weak point

5- The mere comparison made for the same patients before surgery and at least 10 years after surgery is invalid as the risk of CVD increases with age.

**Do you want your identity to be public for this peer review?** For information about this choice, including consent withdrawal, please see our Privacy Policy

Reviewer #3: No

Reviewer #4: No

---

## [Author Response · Author response to Decision Letter 2]

8 Oct 2025

Oct 6th, 2025

Agata Rysinska

PONE-D-24-55835R1

Asymptomatic Osteolysis as a Risk Factor for Cardiovascular Disease After Total Hip Arthroplasty: A retrospective cohort study

Response to Reviewer

Dear Nan Jiang,

Thank you for giving us the opportunity to submit a revised draft of the “Asymptomatic Osteolysis as a Risk Factor for Cardiovascular Disease After Total Hip Arthroplasty:

A retrospective cohort study” manuscript for publication in the PLOS ONE. We appreciate the time and effort that you and the reviewers dedicated to providing feedback on our manuscript and are grateful for the insightful comments on and valuable improvements to our paper. Please see below our comments and answers.

Reviewer #3: I am honored to have the opportunity to review this first study investigating the incidence of CVD in patients with or without asymptomatic osteolysis.

1. Inclusion Criteria for Study Population The manuscript states that patients treated with uncemented THA with metal-on-polyethylene articulation were considered eligible for inclusion. Were ceramic-on-polyethylene and other bearing combinations excluded? If so, what was the rationale for this exclusion? Similarly, what was the reasoning behind excluding cemented cups and cemented stems?

Response: Thank you for this interesting question.

Ceramic heads tend to produce less polyethylene wear than metal heads-less biological provocation to activate macrophages, leading to bone resorption, osteolysis and possibly aseptic loosening. Cemented stems and cups were excluded since cementless components historically had higher rates of cup revision for aseptic loosening. Also uncemented cups had a larger rate of visible periacetabular osteolysis.

2. Exclusion of Bisphosphonate Users Could you please clarify the rationale for excluding patients who were using bisphosphonates?

Response: Bisphosphonates suppress osteoclast mediated bone resorption and could therefore be a potent confounder for development of periprosthetic osteolysis.

3. Baseline Characteristics and Statistical Adjustment The results indicate that "the proportion of patients with previous myocardial infarction was higher in the osteolysis group." Was this baseline imbalance ultimately adjusted for in the final analysis? Is the adjusted hazard ratio of 1.6 (95% CI, 0.5-5.8) the value after adjusting for this and other confounders? I apologize for my limited comprehension, but I would appreciate clarification on this point.

Response: Yes. The adjusted hazard ratio of 1.6 (95% CI 0.5–5.8) is the result after adjusting for age, sex, and pre-existing cardiovascular disease (including prior myocardial infarction).

4. Sample Size and Study Conclusions The study found no significant differences between groups in inflammatory markers (hs-CRP, leukocyte count), lipid levels, or frequency of ECG abnormalities, and the authors attributed this to insufficient sample size. If this is indeed the case, should publication be deferred until a larger sample size can be obtained? I believe there is a substantial possibility that the true conclusion might be that asymptomatic osteolysis is not associated with CVD development. This alternative interpretation should be acknowledged and discussed. Furthermore, I suggest that the value of this paper would be enhanced by addressing how large a sample size would be needed to definitively conclude "no association" and providing your expert opinion on this matter. Such discussion would provide important guidance for future research design and help readers better interpret the clinical significance of these findings.

Response: We performed a post-hoc feasibility calculation for a time-to-event analysis using Schoenfeld’s method. Assuming a two-sided α=0.05, power=80%, exposure prevalence of 24% (osteolysis), and a HR of 1.6, the required number of outcome events would be approximately 195. Translating events to total sample size depends on the anticipated cumulative event rate over follow-up: with event rates of 12%, 15%, and 20%, the corresponding total sample sizes would be ~1,620, ~1,300, and ~974 participants, respectively. We have added this fact to the discussion.

Reviewer #4: I appreciate the effort made by the authors in this interesting study As an observational retrospective study there are some flaws

1:The definite association between periprosthetic osteolysis and the resulted inflammatory response that causes CVS and cardiac disease has not been proven.

Response: Thank You for this true remark.

This was the material that was possible to do research on from Swedish Hip Arthroplasty Register. To prove the link, it would be needed to show temporality, biological plausibility, replication and that inflammation mediates the effect. Research would need to combine different types of research. For example:

A: Mechanistic lab work (in vitro/ animal models) to show that wear particles cause inflammation that causes vascular damage.

B: Large human observational studies (prospective cohorts) to show association and temporality.

C: Interventional evidence (anti-inflammatory or antiparticle strategies) to show reduction in both inflammation/osteolysis and cardiovascular outcomes.

D: Causal interference methods to strengthen casual claims.

2: The multifactorial etiology of cardiovascular disease makes an obvious difficulty of establishing correlation between osteolysis and CVD.

Response:

For several reasons, for example methodological, biological and epidemiological. Both conditions have shared risk factors and therefore confounding. This makes it difficult to establish correlation between periprosthetic osteolysis and cardiovascular disease. Both are common in older populations. Age, obesity, sedentary lifestyle-all increased risk of both. This makes it difficult to explain if osteolysis itself is associated with CVD or if both conditions share similar risk profile. Even with statistical adjustments residual confounding still often remains.

Another factor is that there are different primary systems and mechanisms. Periprosthetic osteolysis is a localized bone pathology around prosthetic implants driven by mainly inflammatory responses to wear particles (debris from polyethylene, metal or ceramic).

CVD on the other hand is a systemic vascular disease influenced by endothelial dysfunction due to inter alia hypertension, inflammation and endothelial dysfunction.

3: It is very difficult to find matched patients’ characteristics in each group with many variables that will affect the result.

Response:

Temporality and timing: periprosthetic osteolysis often develops many years after implantation, and it is a subclinical condition without pain, while CVD can predate or develop in parallel.

Establishing weather periprosthetic osteolysis precedes or contributes to CVD requires longitudinal follow-up studies with repeated imaging and outcome capture which is expensive and logistically difficult. Imaging requires CT analysis, not a routine method.

CVD is defined in different ways (clinical events, imaging, biomarkers), misclassification of either exposure or outcome weakens correlation.

4: The small number of patients studied is another weak point.

Response: Revision surgery for osteolysis is relatively uncommon compared with primary total hip or knee surgeries due to osteoarthritis. Large cohorts and long follow up time are needed to capture both exposures and outcomes which makes studies costly and underpowered.

5: The mere comparison made for the same patients before surgery and at least 10 years after surgery is invalid as the risk of CVD increases with age.

Response: Thank You for a very intriguing question.

Aging is an independent risk factor. With increasing age, the risk of CVD rises regardless of whether they had THA surgery. Time itself is a confounder, during the 10+years other factors as lifestyle changes, new comorbidities and/or medications could affect the CVD risk

Comparing “before THA” when patients are younger 10+years later doesn´t isolate the effect of THA, it just shows that the risk for CVD increases with higher age.

Without controlling for aging and confounding by time it is hard to explain weather the risk for CVD is because of the THA or due to natural ageing and disease progression.

Sincerely

Agata Rysinska

---

## [Decision Letter · Decision Letter 2]

20 Oct 2025

Dear Dr. Rysinska,

Thank you for submitting your manuscript to PLOS ONE. After careful consideration, we feel that it has merit but does not fully meet PLOS ONE’s publication criteria as it currently stands. Therefore, we invite you to submit a revised version of the manuscript that addresses the points raised during the review process.

We look forward to receiving your revised manuscript.

Kind regards,

Nan Jiang

Academic Editor

PLOS ONE

Journal Requirements:

Reviewers' comments:

Reviewer's Responses to Questions

**Comments to the Author**

Reviewer #3: (No Response)

2. Is the manuscript technically sound, and do the data support the conclusions?

Reviewer #3: Yes

3. Has the statistical analysis been performed appropriately and rigorously?

Reviewer #3: Yes

4. Have the authors made all data underlying the findings in their manuscript fully available?

Reviewer #3: Yes

5. Is the manuscript presented in an intelligible fashion and written in standard English?

Reviewer #3: Yes

Reviewer #3: It is a great honor to have the opportunity to review this valuable manuscript.

The authors state that they "previously showed an increased long-term risk of cardiovascular-related mortality in patients treated with THA [16], and an increased relative risk of cerebrovascular events (CVE) in patients with osteoarthritis who received THA and subsequently underwent revision surgery due to loosening of the prosthesis [17]. Similarly, cardiovascular mortality following total knee arthroplasty exceeds that of the general population after 10 years [18]."

I was completely unaware of these findings and found them highly informative. However, I am disappointed that the direct relationship could not be statistically demonstrated in this study. I would encourage the authors to consider whether statistical significance could be achieved through increased sample size or more rigorous matching.

The authors also state that "the absence of notable differences in this study does not conclusively negate the possibility of osteolysis serving as a causal factor for CVD, as the chronic low-grade inflammation in osteolysis could hypothetically contribute to development of atherosclerosis." In this regard, I wonder whether it would be possible to examine the association between atherosclerotic burden and the presence or absence of osteolysis.

The manuscript would benefit from including representative CT images: at least one demonstrating periacetabular osteolysis and one without osteolysis.

Finally, regardless of whether an association between osteolysis and CVD is confirmed, the manuscript would provide greater clinical value if the authors could offer recommendations for orthopedic surgeons regarding strategies to reduce CVD risk in THA patients during long-term follow-up.

**Do you want your identity to be public for this peer review?** For information about this choice, including consent withdrawal, please see our Privacy Policy

Reviewer #3: No

---

## [Author Response · Author response to Decision Letter 3]

12 Nov 2025

Nov 10th, 2025

Agata Rysinska

PONE-D-24-55835R1

Asymptomatic Osteolysis as a Risk Factor for Cardiovascular Disease After Total Hip Arthroplasty: A retrospective cohort study

Response to Reviewer

Dear Nan Jiang,

Thank you for giving us the opportunity to submit a revised draft of the “Asymptomatic Osteolysis as a Risk Factor for Cardiovascular Disease After Total Hip Arthroplasty:

A retrospective cohort study” manuscript for publication in the PLOS ONE. We appreciate the time and effort that you and the reviewers dedicated to providing feedback on our manuscript and are grateful for the insightful comments on and valuable improvements to our paper. Please see below our comments and answers.

6. Review Comments to the Author

Reviewer #3: It is a great honor to have the opportunity to review this valuable manuscript.

The authors state that they "previously showed an increased long-term risk of cardiovascular-related mortality in patients treated with THA [16], and an increased relative risk of cerebrovascular events (CVE) in patients with osteoarthritis who received THA and subsequently underwent revision surgery due to loosening of the prosthesis [17]. Similarly, cardiovascular mortality following total knee arthroplasty exceeds that of the general population after 10 years [18]."

I was completely unaware of these findings and found them highly informative. However, I am disappointed that the direct relationship could not be statistically demonstrated in this study. I would encourage the authors to consider whether statistical significance could be achieved through increased sample size or more rigorous matching.

Action:

An increased sample size would improve statistical analysis, ton increase the current study is ouside of the scope of this study. We do however believe that our results could be useful in the building of future hypotheses and meta-analyses. A more rigorous matching on variables that plausibly can affect both conditions.

Adults over 50 years undergoing total hip or knee arthroplasty surgery with baseline imaging. Exclusion criteria would be pre-existing advanced CVD, known active infection and immunosuppression that will confound inflammatory markers.

Baseline cardiovascular risk such as history of CVD, hypertension, diabetes, hyperlipidemia, smoking and BMI.

Medications, baseline statins, antihypertensives, antiplatelets and anticoagulants.

Multicenter study can provide implant heterogeneity. Implant type, material, fixation methods (exact match or cluster).

The authors also state that "the absence of notable differences in this study does not conclusively negate the possibility of osteolysis serving as a causal factor for CVD, as the chronic low-grade inflammation in osteolysis could hypothetically contribute to development of atherosclerosis." In this regard, I wonder whether it would be possible to examine the association between atherosclerotic burden and the presence or absence of osteolysis.

Action:

To establish association between atherosclerosis and periprosthetic osteolysis is difficult for many reasons: scientific, methodological and biological.

The primary systems and mechanisms differ.

Atherosclerosis is a systemic vascular disease influenced by lipid metabolism, endothelial dysfunction, hypertension and inflammation.

Periprosthetic osteolysis is bone pathology locally around prosthetic implants driven by inflammatory responses to wear particles (polyethylene, metal or ceramic debris).

Both involve inflammation, both are distinct and overlap only partially.

There are similar confounding mechanisms, due to similar age-related risk factors such a age, obesity, diabetes, smoking, inactivity etc. The shared risk factors make it difficult to determine if there is a direct relationship between atherosclerosis and periprosthetic osteolysis or if both conditions occur more often in older patients with comorbidities.

Another challenge is measurement and detection. Both conditions can occur silently for many years before being detected, therefore this mismatch in time and diagnostic variability make longitudinal correlation difficult. Standardized biomarkers or imaging endpoints are lacking for both processes currently. Cardiovascular studies don´t collect data on implant-related bone changes, and similarly: studies on periprosthetic osteolysis focus on orthopedic measurable outcomes-not systemic health. The heterogeneity in study designs limits the ability to establish association.

Prospective human studies to test this association would require large, long-term cohorts, costly and difficult.

The manuscript would benefit from including representative CT images: at least one demonstrating periacetabular osteolysis and one without osteolysis.

Action:

Due to the restrictions of the ethical permit, we are unable to provide the requested CT images.

Finally, regardless of whether an association between osteolysis and CVD is confirmed, the manuscript would provide greater clinical value if the authors could offer recommendations for orthopedic surgeons regarding strategies to reduce CVD risk in THA patients during long-term follow-up.

Action:

Should this association be proven, surgeons should take care to include and optimize specific follow up surveillance protocols regarding CVD in these patient populations. However, these protocols should be the subject of interdisciplinary collaboration between orthopaedic surgeons, cardiologists and radiologists. We have adjusted the manuscript accordingly.

Sincerely

Agata Rysinska

---

## [Decision Letter · Decision Letter 3]

25 Nov 2025

Asymptomatic Osteolysis as a Risk Factor for Cardiovascular Disease After Total Hip Arthroplasty:

A retrospective cohort study

PONE-D-24-55835R3

Dear Dr. Rysinska,

We’re pleased to inform you that your manuscript has been judged scientifically suitable for publication and will be formally accepted for publication once it meets all outstanding technical requirements.

Kind regards,

Nan Jiang

Academic Editor

PLOS ONE

Reviewers' comments:

Reviewer's Responses to Questions

**Comments to the Author**

Reviewer #3: All comments have been addressed

2. Is the manuscript technically sound, and do the data support the conclusions?

Reviewer #3: (No Response)

3. Has the statistical analysis been performed appropriately and rigorously?

Reviewer #3: (No Response)

4. Have the authors made all data underlying the findings in their manuscript fully available?

Reviewer #3: (No Response)

5. Is the manuscript presented in an intelligible fashion and written in standard English?

Reviewer #3: (No Response)

Reviewer #3: (No Response)

**Do you want your identity to be public for this peer review?** For information about this choice, including consent withdrawal, please see our Privacy Policy

Reviewer #3: No

---

## [Editor Report · Acceptance letter]

PONE-D-24-55835R3

PLOS One

Dear Dr. Rysinska,

I'm pleased to inform you that your manuscript has been deemed suitable for publication in PLOS One. Congratulations! Your manuscript is now being handed over to our production team.

Kind regards,

on behalf of

Dr. Nan Jiang

Academic Editor

PLOS One